# Extremely confined gap plasmon modes: when nonlocality matters

Sergejs Boroviks [1,2,3], Zhan-Hong Lin [2], Vladimir A. Zenin [1], Mario Ziegler [2], Andrea Dellith[2], P. A. D. Gonçalves [1], Christian Wolff [1], Sergey I. Bozhevolnyi [1,4], Jer-Shing Huang [2,5,6,7] & N. Asger Mortensen [1,4 ✉]

Historically, the field of plasmonics has been relying on the framework of classical electrodynamics, with the local-response approximation of material response being applied even when dealing with nanoscale metallic structures. However, when the confinement of electromagnetic radiation approaches atomic scales, mesoscopic effects are anticipated to become observable, e.g., those associated with the nonlocal electrodynamic surface response of the electron gas. Here, we investigate nonlocal effects in propagating gap surface plasmon modes in ultrathin metal–dielectric–metal planar waveguides, exploiting monocrystalline gold flakes separated by atomic-layer-deposited aluminum oxide. We use scanning near-field optical microscopy to directly access the near-field of such confined gap plasmon modes and measure their dispersion relation via their complex-valued propagation constants. We compare our experimental findings with the predictions of the generalized nonlocal optical response theory to unveil signatures of nonlocal damping, which becomes appreciable for few-nanometer-sized dielectric gaps.

[1] Center for Nano Optics, University of Southern Denmark, Campusvej 55, DK-5230 Odense M, Denmark. [2] Leibniz Institute of Photonic Technology, Albert-Einstein Straße 9, 07745 Jena, Germany. [3] Nanophotonics and Metrology Laboratory, Swiss Federal Institute of Technology Lausanne (EPFL), Station 11, CH 1015 Lausanne, Switzerland. [4] Danish Institute for Advanced Study, University of Southern Denmark, Campusvej 55, DK-5230 Odense M, Denmark. [5] Institute of Physical Chemistry and Abbe Center of Photonics, Friedrich-Schiller-Universität Jena, 07743 Jena, Germany. [6] Research Center for Applied Sciences, Academia Sinica, 128 Sec. 2, Academia Road, Nankang District 11529 Taipei, Taiwan. [7] Department of Electrophysics, National Yang Ming Chiao Tung University, 1001 University Road, 30010 Hsinchu, Taiwan. ✉email: asger@mailaps.org

One of the appealing features of plasmonics—the possibility to squeeze light beyond the diffraction limit and guide electromagnetic energy in waveguides with sub-wavelength dimensions[1–3]—has remained at the heart of the nanophotonics community for several decades[4]. Such tight focusing is only possible due to unique properties of the surface-plasmon polariton (SPP), being a collective oscillation of the free charge-carriers in metals coupled with light fields[5]. Various configurations, designs, and materials for plasmonic systems have been studied from both fundamental and applied perspectives, attempting to realize a diversity of functional devices, ranging from plasmonic integrated circuitry[6] to metasurface-based flat optical components[7].

In particular, many devices are based on metal–dielectric–metal (MDM) heterostructures, which support gap surface plasmon (GSP) modes[8]. GSPs can lead to substantial electric-field enhancements inside the dielectric gap, which can strongly enhance linear and nonlinear optical processes[9], and may be exploited for various applications of plasmon-enhanced light–matter interactions, e.g., surface-enhanced Raman spectroscopy (SERS)[10] or Purcell enhancement of the emission rate of single-photon sources[11,12]. Furthermore, MDM heterostructures may be engineered to achieve larger mode propagation lengths with better field confinements when compared with other SPP modes[13,14]. The GSP modes supported by such waveguides were extensively studied by various experimental methods, including far-field techniques[15,16] and scanning near-field optical microscopy (SNOM)[17,18].

The above-mentioned ambitions, along with the promise of realizing plasmonic waveguides for integration, or even replacement of conventional photonic waveguides, largely rely on the aspiration of significantly mitigating Ohmic losses inherent to metals[19,20]. Indeed, some progress in the development of traditional and alternative plasmonic materials has been made[21,22]. Moreover, with advances in colloidal synthesis methods of traditional plasmonic materials, it became possible to grow monocrystalline gold (Au) flakes with high aspect ratio—few tens of nanometers in thickness and up to several hundreds of microns in lateral size[23,24]. Along with the improvement in nanofabrication techniques, such as focused ion beam (FIB) milling, these progresses allowed the fabrication of plasmonic nanocircuitry[25,26] and plasmonic nanoantennas[27,28] with superior quality. Furthermore, monocrystalline Au flakes present as a "playground" material platform for experimental studies of fundamental aspects of plasmonics[29–36].

Thus far, most of the novel aspects in plasmonics have emerged and developed from considerations rooted in classical electrodynamics and the local-response approximation (LRA) for the interaction of light with the free-electrons in metals[5]. However, in recent years there has been increasing attention to quantum plasmonics[37–39] and the importance of quantum corrections to classical electrodynamics in plasmonic nanostructures[40–47], including nonlocal effects impacting GSP modes in MDM structures with ultranarrow gaps[48,49] and hyperbolic metamaterials[50]. This has contributed to a general recognition that in addition to bulk losses in plasmonic metals, there is an additional contribution in metallic surfaces associated with nonlocal effects[20,41,45].

The dominant microscopic mechanism contributing to nonlocal losses is Landau damping[51], which becomes especially pronounced at large plasmonic mode propagation vectors, and thus manifesting also in MDM structures with very small dielectric gaps. While ab initio approaches in principle account for such quantum nonlocal effects[52], quantum corrections to the LRA may also be explored semi-analytically using either hydrodynamic models or a surface-response formalism (SRF)[45]. In

particular, the use of Feibelman $d$-parameters[53] has recently been revived in the context of nanogap structures and related plasmonic phenomena[44,54–59]. At the same time, there has been modest progress in experimental investigations of extremely confined propagating GSP modes as far as nonlocal corrections to the GSP dispersion in nanometer-sized gaps are concerned. The most relevant characterization of GSP modes is associated with observations of large splitting of symmetric and antisymmetric eigenmodes in side-by-side aligned single-crystalline Au nanorod dimers with atomically defined gaps reaching ~5 Å[60]. While the experimentally measured resonance wavelengths could be accounted for without involving quantum nonlocal effects, experimentally measured resonance quality factors for GSP-based resonances were noted to be significantly lower than those predicted by the classical, local-response theory[60]. This qualitative observation tallies well with the analysis of the Landau damping influence on the GSP dispersion for nanometer-sized gaps that revealed little difference between local and nonlocal considerations of the real part of the GSP propagation constant, while showing progressively strong increase of the GSP propagation loss for gaps below 10 nm[51]. Given the importance of GSP-based configurations for a wide range of plasmon-mediated light–matter interactions[9], it is crucial to experimentally establish a benchmark for the nonlocal (i.e., Landau or surface collision) damping associated with extremely confined GSP modes.

In this work, we present an experimental study of extremely confined GSP modes in MDM structures fabricated out of monocrystalline Au flakes and atomic-layer-deposited (ALD) ultrathin aluminum oxide ($Al_2O_3$) films. The use of crystalline metal and high-quality dielectric material is crucial for reducing bulk and surface-roughness related losses to the minimum, thus opening a doorway to explore nonlocality[48]. The concept of the experiment is schematically illustrated in Fig. 1. Using a scattering-type SNOM (s-SNOM)[61], we obtain near-field (NF) maps of propagating GSP modes, which exhibit, to the best of our knowledge, a record-high experimentally demonstrated effective-mode index, reaching values of approximately 6.2 in the case of a ~2 nm-thick gap at $\lambda_0 = 1550$ nm excitation wavelength. Further analysis of the experimental data from a range of samples suggests signatures of gap-dependent nonclassical damping, being especially pronounced in samples with dielectric gap thicknesses

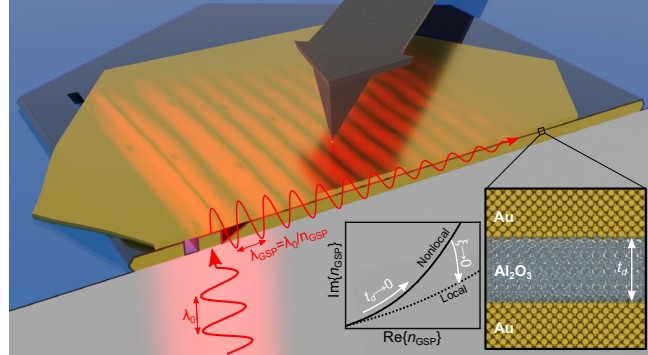

**Fig. 1 Schematic illustration of the experiment.** The red curves schematically illustrate the electric-field profile of the excitation light (bottom left) and of the propagating GSP mode (in the sample) with the group-index $n_{GSP}$ (corresponding to a GSP-wavelength $\lambda_{GSP}$ at the free-space wavelength $\lambda_0$). Insets show the parametric plot (varying the gap-size, $t_d$) of the GSP dispersion trends as the nonlocal length scale $\xi \longrightarrow 0$, while at the bottom right it is shown a close-up of the MDM waveguide comprised of the monocrystalline gold (Au) flakes separated by a thin dielectric gap formed by atomic-layer deposition of aluminum oxide ($Al_2O_3$).

smaller than 5 nm. We show that such observations are compatible with a nonclassical interpretation in terms of the so-called generalized nonlocal optical response (GNOR) theory[41], and provide an estimate of the diffusion constant that accounts for nonlocal damping. Incidentally, although the microscopic origin of the diffusion constant in GNOR represents carrier-scattering, it can also impersonate other sources of nonlocal damping (e.g., Landau damping) in a phenomenological fashion.

## Results

**Quantum nonlocal corrections to the GSP dispersion relation.** The dispersion relation of the fundamental GSP mode in a MDM structure is generally given by[48]

$$\tanh\left(\frac{\kappa_{\mathrm{d}} t_{\mathrm{d}}}{2}\right) = -\frac{\kappa_{\mathrm{m}}\varepsilon_{\mathrm{d}}}{\kappa_{\mathrm{d}}\varepsilon_{\mathrm{m}}}\left(1 + \delta_{\mathrm{nl}}\right), \qquad (1\mathrm{a})$$

which reduces to the LRA expression of Economou[5] in the absence of nonlocal corrections (i.e., $\delta_{\mathrm{nl}} \to 0$). Here, $t_{\mathrm{d}}$ is the thickness of the dielectric gap, whereas the out-of-plane components of the wavevectors are defined as

$$\kappa_j = \sqrt{q^2 - \varepsilon_j k_0^2} \quad \text{for} \quad j \in \{\mathrm{d}, \mathrm{m}\}, \qquad (1\mathrm{b})$$

where $q$ is the GSP propagation constant, $k_0 \equiv \omega/c = 2\pi/\lambda_0$ is the free-space wavevector, and $\varepsilon_{\mathrm{d}} \equiv \varepsilon_{\mathrm{d}}(\omega)$ and $\varepsilon_{\mathrm{m}} \equiv \varepsilon_{\mathrm{m}}(\omega)$ are the relative permittivities ($q$-independent) of the dielectric and the metal, respectively.

Within the nonlocal hydrodynamic formalism[41,42,45], the nonlocal response introduces $q$-dependence in the material response (giving rise to a nonlocal response in the corresponding real-space representation) of the form[48]

$$\delta_{\mathrm{nl}}(q, \omega) = \frac{q^2}{\kappa_{\mathrm{nl}}\kappa_{\mathrm{m}}}\frac{\varepsilon_{\mathrm{m}} - \varepsilon_\infty}{\varepsilon_\infty}, \qquad (2\mathrm{a})$$

$$\kappa_{\mathrm{nl}} = \sqrt{q^2 - \frac{1}{\xi^2}\frac{\varepsilon_{\mathrm{m}}}{\varepsilon_\infty}}. \qquad (2\mathrm{b})$$

The nonlocal wavevector $\kappa_{\mathrm{nl}} \equiv \kappa_{\mathrm{nl}}(q, \omega)$ enters in the hydrodynamic model of plasmonics as an additional longitudinal wave[42], with the nonlocal length scale $\xi$ given by[42,45].

$$\xi^2 = \frac{(3/5)v_{\mathrm{F}}^2}{\omega(\omega + \mathrm{i}\gamma)} + \frac{\mathcal{D}}{\mathrm{i}\omega}. \qquad (2\mathrm{c})$$

Here, the first term originates from the Thomas–Fermi theory of metals[42], with $v_{\mathrm{F}}$ being the Fermi velocity. The second term is an addition from GNOR model, with $\mathcal{D}$ denoting the diffusion constant that embodies the nonlocal damping. Finally, $\varepsilon_\infty \equiv \varepsilon_\infty(\omega)$ is a heuristic frequency-dependent parameter originating from the Drude LRA model $[\varepsilon_{\mathrm{m}} \equiv \varepsilon_\infty - \omega_{\mathrm{p}}^2/(\omega^2 + \mathrm{i}\gamma\omega)]$ that takes into account polarization due to the presence of positively charged atomic ions and interband transitions in the background of the quasi-free electron gas.

While the quantum-corrected dispersion relation (1a) originates from a hydrodynamic treatment of the nonlocal response of metals (with $\mathrm{Re}\{\delta_{\mathrm{nl}}\} \propto v_{\mathrm{F}}$ and $\mathrm{Im}\{\delta_{\mathrm{nl}}\} \propto \sqrt{\mathcal{D}}$)[42], we emphasize that in a SRF[45,53], it can equally well be expressed in terms of the Feibelman parameter for the centroid of the induced charge[53], i.e., $\delta_{\mathrm{nl}} \propto d_\perp$ (see Supplementary Section S10). For convenience, we will use the effective-mode index $n_{\mathrm{GSP}} \equiv q/k_0 = \mathrm{Re}\{n_{\mathrm{GSP}}\} + \mathrm{i} \cdot \mathrm{Im}\{n_{\mathrm{GSP}}\}$ and thus discuss all our results in terms of this dimensionless and complex-valued quantity. Naturally, the stronger the confinement of electromagnetic field is, the larger $\mathrm{Re}\{n_{\mathrm{GSP}}\}$. In the spirit of the above treatment, where $\delta_{\mathrm{nl}} \ll 1$, nonlocality is expected to make only a small correction, which can nevertheless become sizable for large GSP wavevectors

(herein promoted by small, nanometer-scale dielectric gap thicknesses, see schematic inset in Fig. 1), i.e., for large $\mathrm{Re}\{n_{\mathrm{GSP}}\}$. Intuitively, $\mathrm{Re}\{n_{\mathrm{GSP}}\} \gg \mathrm{Im}\{n_{\mathrm{GSP}}\}$ is required for rendering quantum nonlocal effects experimentally observable. To enter this regime, the surface-response function $\mathrm{Re}\{d_\perp\}$ should not be negligible in comparison to $t_{\mathrm{d}}$, thus calling for nanofabrication techniques that can controllably realize MDM structures with sub-10-nanometer gaps. For sub-nanometric gaps with $t_{\mathrm{d}} \lesssim |\mathrm{Re}\{d_\perp\}|$, additional quantum mechanical effects (e.g., tunneling, electronic spill-out) may be needed to be incoporated as well[38,55].

**Sample design and fabrication.** We experimentally study the dependence of the GSP spectral features on the thickness of the dielectric gap $t_{\mathrm{d}}$. To that end, we have fabricated five different planar MDM structures with varying $t_{\mathrm{d}}$ (nominally 2, 3, 5, 10, and 20 nm), along with tailored waveguide couplers. As previously mentioned, in order to explore potential signatures of nonlocal effects, it is essential to reduce as much as possible all losses of intrinsic origin (i.e., classical, bulk losses), but also those related to fabrication imperfections, such as surface roughness, contamination of metal and dielectric materials with impurities, etc. Therefore, we have utilized monocrystalline Au flakes as metal layers, whereas for the dielectric material in the core of the MDM structure we have employed plasma-assisted ALD of $Al_2O_3$ layers which allows controlled growth of homogeneous dielectric layers with approximately 2.2 Å precision (see Methods and Supplementary Fig. S1 for details).

Figure 2 shows optical (a–d) and scanning electron (e, f) micrographs of a fabricated sample with a 3 nm dielectric gap, also revealing the FIB-milled tapered waveguide coupler. This element of the sample design and fabrication is of particular importance, since due to the large wavevector mismatch between free-space light and confined GSP modes, excitation of the latter with a Gaussian laser beam is not efficient. As such, it is necessary to provide a compact and adequate coupling mechanism, which is critical to obtain sufficiently strong signal in SNOM measurements. Due to the short GSP propagation length ($L_{\mathrm{GSP}} \equiv [2\mathrm{Im}\{q\}]^{-1}$ is less than 0.5 $\mu$m for a 2-nm-thick dielectric gap), a typical grating coupler schemes[62] become unsuitable. The periodicity of the grating $\Lambda$ required by the phase-matching condition $q = k_0 + k_{\mathrm{grating}} = k_0 \pm m2\pi/\Lambda$ (with $m$ being an integer) even for a grating with just 3 periods, is comparable with the GSP propagation length. Therefore, in order to improve the coupling efficiency while maintaining a compact device, we exploited a tapered waveguide coupler design[63] and optimized its geometrical parameters for each dielectric gap thickness (see Methods and Supplementary Section S9 for details).

Another important aspect of the design and fabrication of the MDM waveguides concerns the thickness of the upper gold flake, $t_{\mathrm{u}}$ (see sketch in Supplementary Fig. S13a). On the one hand, this upper Au layer needs to be sufficiently thin to make the NF of the ultraconfined GSP modes accessible to the s-SNOM tip, which scatters only weak evanescent tails of the mode penetrating through the top flake. On the other hand, $t_{\mathrm{u}}$ should be large enough to avoid significant modification of the GSP mode by undesired hybridization with the bare, single-interface SPP mode at the top air–gold interface. In both cases, the characteristic length-scale is the skin depth (~30 nm), making it a challenging task to fulfill both requirements. However, after thorough work on noise suppression in our SNOM, we managed to obtain near-field maps with sufficient signal-to-noise ratio for an upper-flake thickness of ~50 nm, for which the hybridization between the fundamental GSP and the single-interface SPP mode is negligible, while the assignment of a local thickness-independent $\varepsilon_{\mathrm{m}}$ is also justified[59,64]. Furthermore, atomic flatness of the monocrystalline

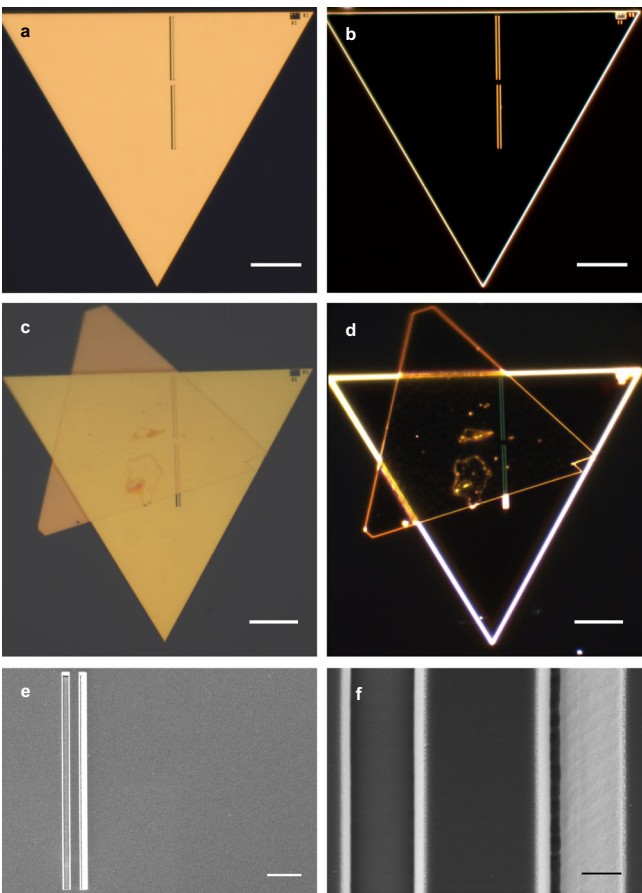

**Fig. 2 Micrographs of the fabricated sample. a** Bright- and (**b**) dark-field optical images of the sample with a 3 nm $Al_2O_3$ layer before the transfer of the top flake. **c** Bright- and (**d**) dark-field optical images of the sample after the transfer of the top flake. **e** SEM image of the flake during intermediate fabrication step and (**f**) close-up image of the FIB milled coupling element. Scale bars in panels (**a–d**) correspond to 10 μm, in panel (**e**) to 2 μm and in panel (**f**) to 50 nm.

Au flakes allows to reduce the noise induced by surface roughness in near-field measurement, which is an important aspect for such low-signal measurements.

**Near-field measurements.** Using the capabilities of our s-SNOM setup to measure both amplitude and phase, as illustrated in Fig. 3a (detailed descriptions of the setup can be found in the Methods section), we have obtained complex NF maps showing the propagating GSP modes in all the fabricated samples. As an example, Fig. 3b and c show pseudo-color images of the electric NF amplitude $|E_{NF}|$ and its real part $Re\{E_{NF}\}$, respectively, for the sample with $t_d = 3$ nm (NF maps of all other samples are provided in Supplementary Fig. S3). One-dimensional Fourier-transformation of the recorded NF maps along the GSP propagation direction ($x \rightarrow k_x$, as illustrated in Fig. 3d via the normalized absolute value of the transformed image) and averaging of the $E(k_x)$ spectrum along the $y$-axis direction, allows us to extract the real part of the GSP effective-mode index. As exhibited in Fig. 3e, the NF spectrum has peaks at two spatial frequencies, which correspond to two distinct propagating modes. The first one, with an effective-index of approximately unity, can be attributed to free-space light at an oblique incidence or to SPP modes in the topmost air–gold interface. The other contribution corresponds to the GSP mode and manifests as a peak at an effective-index slightly exceeding $Re\{n_{GSP}\} \approx 5.1$. We note the

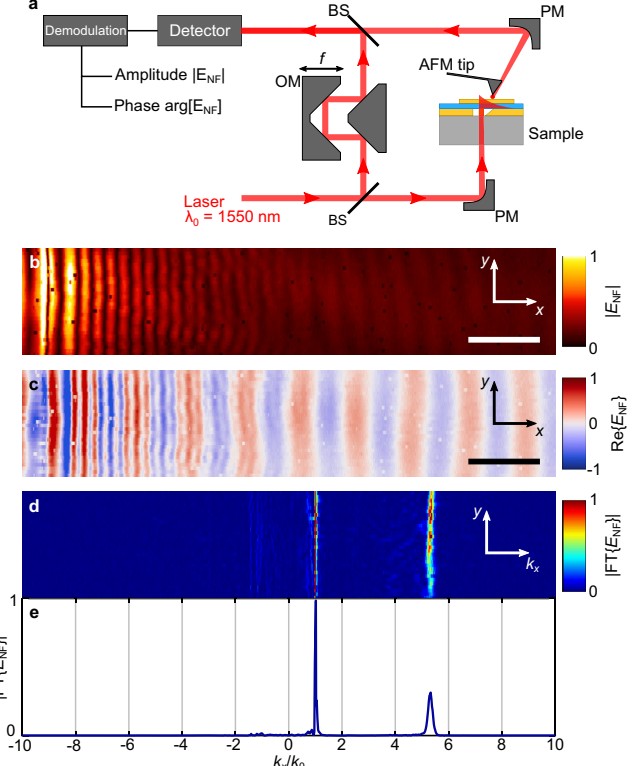

**Fig. 3 Scanning near-field optical microscopy measurements. a** Schematic illustration of the s-SNOM setup (see "Methods" for details). Pseudo-color images of (**b**) amplitude and (**c**) real-part of the detected near-field (NF) signal for the sample with 3 nm dielectric gap at $\lambda_0 = 1550$ nm excitation wavelength (scale bars: 2 μm). **d** Amplitude of the Fourier transformed NF map along the propagation coordinate ($x \rightarrow k_x$) and (**e**) its profile (averaged along the $y$-axis).

absence of any significant contribution at a corresponding negative value ($Re\{n_{GSP}\} \approx -5.1$), which could indicate GSP back-scattering ($k_x \rightarrow -k_x$) due to surface roughness, potentially competing with nonlocal effects[65]. Moderate roughness-induced scattering, as well as inhomogeneity of the effective deielectric constant, may result in the broadening of the forward-scattering peak ($k_x \rightarrow k_x + \Delta k_x$, $\Delta k_x \ll k_x$), which would effectively manifest in a slightly increased imaginary part of the wavevector.

Further data post-processing, namely filtering of the NF maps and selecting only spatial frequencies in the vicinity of $n_{GSP}$ in the Fourier domain, allows us to clean up interference with other near-fields as well as to reduce the noise, and retrieve the spatial evolution of the pure GSP mode along the propagation direction. Figure 4a illustrates the results of that procedure, showing how the GSP wavelength is shortened ($Re\{n_{GSP}\}$ increased) and exhibits a faster decay (i.e., increasing $Im\{n_{GSP}\}$) as $t_d$ is reduced from $t_d = 20$ nm down to 2 nm. The propagation length of the GSP modes can be estimated by fitting an exponential envelope to the $|E_{NF}|$ field. Further details about the NF map processing can be found in Supplementary Section S4.

**Fit of the experimental results to GNOR model.** The parametric plot in Fig. 4b summarizes the experimental results (square data-points with error bars) and contrasts them against classical LRA calculations (dashed curve) as well as to those based on the GNOR model (solid curve). In both calculations, we used experimentally obtained material parameters at $\lambda_0 = 1550$ nm from literature, namely, $\varepsilon_m = -106.62 + 6.1257i$ for

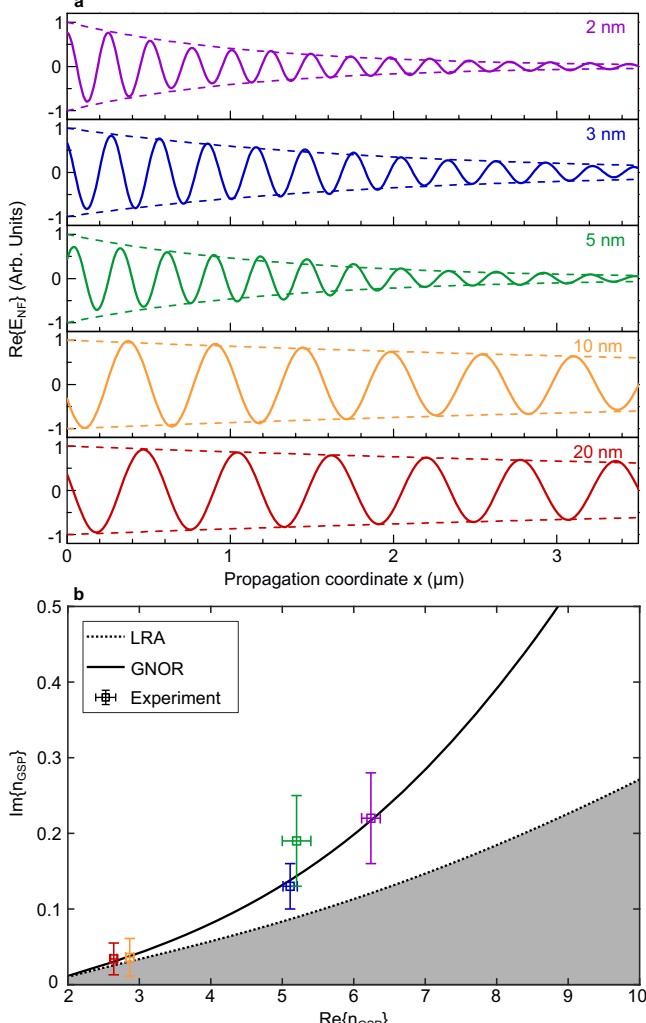

**Fig. 4 Analysis of the experimental and theoretical results. a** Real part (solid curves) and exponential envelopes (dashed curves) of the GSP mode profiles extracted from s-SNOM measurements for the five samples with different gap thickness (see labels). **b** Parametric plot of the effective-mode index $n_{GSP}$ (at the excitation wavelength $\lambda_0 = 1550$ nm) for varying dielectric gap thickness: calculated using LRA (dashed curve), GNOR model (solid curve) and experimentally obtained data (squares with error bars that represent standard deviation of the measurement). Colors of the indicated points on the curves and experimental data points correspond to 2, 3, 5, 10, and 20 nm gap thicknesses, as in panel (a).

monocrystalline Au from Olmon et al.[66] and $\varepsilon_d = 2.657$ for $Al_2O_3$ from Boidin et al.[67]. The Drude model parameters which enter the nonlocal correction factor, $\delta_{nl}$, were obtained by fitting the experimental data from Olmon et al. in the 1000–2000 nm wavelength range, yielding $\hbar\omega_p = 8.29$ eV, $\hbar\gamma = 47.72$ meV and $\varepsilon_\infty = 2.61$; and a tabulated value for gold's Fermi velocity $v_F = 1.4 \times 10^6$ ms$^{-1}$ was used[68]. Even though the slope of the LRA curve is sensitive to the exact value of $\varepsilon_m$ (see Supplementary Fig. S7), we deliberately invoke data from Olmon et al.[66], being appropriate for descriptions of optical response of monocrystalline Au flakes[23]. The diffusion constant $\mathcal{D}$ is used as the GNOR model's fitting parameter (see details in Supplementary Section S5). By fitting the dispersion relation [Eq. (1a)] to the the imaginary part of the experimental data points (at the corresponding values of Re{$n_{GSP}$}), we obtained an estimate

$\mathcal{D} \simeq 8 \times 10^{-4}$ m$^2$ s$^{-1}$. We emphasize that this procedure does not strictly rely on particular assumptions about the parameters of the dielectric gap, thus allowing a robust extraction of $\mathcal{D}$ even with significant uncertainties in the thickness $t_d$ and effective permittivity $\varepsilon_d$. Further justification of this point is discussed in the following section.

As the LRA represents a minimal model for the light–matter interactions (disregarding nonclassical damping mechanisms), the gray-shaded area of this parametric regime in Fig. 4b would be "prohibited". Within the measurement accuracy of our s-SNOM experiment, all the data points indeed fall outside this regime, except for points corresponding to the large gaps, where quantum nonlocal contributions become negligible and the solid line asymptotically approaches the LRA curve. However, for the smaller gaps the deviations of the experimental data from the LRA curve suggest evidence for including gap-dependent broadening mechanisms in theoretical treatments of the ultraconfined GSP modes, e.g., nonlocal surface-scattering corrections.

## Discussion

Nonlocal corrections to the classical, local-response electrodynamics have traditionally been extracted from optical measurements by confronting experimental data with classical predictions based on the LRA that invoke precise information about morphology of the sample. Typically, this information is obtained via structural characterization techniques, e.g., scanning electron microscopy (SEM), revealing particle or gap sizes and shapes (see, for instance, refs. [69–75]). The success of these approaches to determine nonlocal corrections thus relies on the ability to correlate spectral shifts of resonances with accurate structural information on the morphology, which is no simple task even with state-of-the-art electron microscopy. Other approaches utilized well-controlled spacer materials, which are challenging to work with, such as gaps formed by varying numbers of stacked graphene layers[76,77], while graphene plasmons may also be used on their own to unveil nonlocal quantum effects in nearby metals and to probe surface-response functions[58].

As shown in Fig. 4, our analysis of GSP propagation partially eliminates the above-noted challenge since the experimentally measured $n_{GSP}$ data can be considered in a parametric way (where $t_d$ is varied through different, but otherwise identical devices) and can be confronted with the theory in the same way, i.e., by plotting solutions of Eq. (1a) for varying $t_d$. As such, the dashed curve in Fig. 4b depends only on $\varepsilon_m$ and $\varepsilon_d$, while the solid curve has an additional dependence on $\delta_{nl}$. The solid curve is swept away from the dashed counterpart when one increases $\delta_{nl}$ (effectively by varying complex-valued $\xi$, see inset in Fig. 1). Furthermore, as shown in Supplementary Fig. S8, the slopes of both the LRA and GNOR dispersion curves in this parametric space are nearly independent of $\varepsilon_d$, and insensitive to addition of a thin air void in the gap too (Supplementary Fig. S11). This alleviates the need for extremely accurate quantification of the possible deviations of $\varepsilon_d$ and $t_d$ (as fabricated) from its nominal values (as intended in the initial design).

While we note that the real part of $n_{GSP}$ obtained in experiments is consistently smaller than the predictions of both the LRA and calculations incorporating nonlocal corrections (see Supplementary Fig. S4), this fact does not deny nor contradict our analysis in the parametric way, as explained above. A possible explanation for smaller $n_{GSP}$ could be presence of an air void or an additional dielectric layer between the Au flakes and the ALD film. The former could be present due to poor adhesion or other fabrication imperfection (for example, contamination with small particles that suspend the upper Au flakes slightly above the ALD

layer), while the latter might appear as residuals of the organic molecules from the Au crystal growth solution or deposited conduction layer for FIB milling. We refer to the additional calculations provided in Section S8 of Supplementary Information, which support the plausibility of this explanation, as well as its compatibility with our analysis.

The diffusion constant $\mathcal{D}$, which embodies nonlocal damping, was estimated from a fit to the experimental data (see Section S5 of the Supplementary Information). The obtained value $\mathcal{D} \simeq 8 \times 10^{-4}\,\mathrm{m^2\,s^{-1}}$ is slightly smaller than previously reported experimental value $\mathcal{D} \simeq 8.8 \times 10^{-4}\,\mathrm{m^2\,s^{-1}}$ [75]. In the initial GNOR theory[41], it was consolidated that the diffusion is effectively linked to the surface-scattering properties[78] and that the diffusion constant for air-gold interface is anticipated to be even smaller, on the order of $\mathcal{D} \sim 3 \times 10^{-4}\,\mathrm{m^2\,s^{-1}}$ [79], while dielectric surroundings $(\varepsilon_\mathrm{d} > 1)$ further promotes Landau damping[80,81]. Eventually, the higher diffusion constant revealed by our experiments (as well as ref. [75]) could also indicate some additional surface scattering in our samples, despite our dedicated efforts to eliminate surface roughness through the use of monocrystalline gold and ALD films. Overview and comparison of the previously reported values of $\mathcal{D}$ is provided in Supplementary Table S1 of the Supplementary Information.

We emphasize that we have refrained from any attempt to fit the real part of $\xi$, related to the Fermi velocity $v_\mathrm{F}$, to force nonlocal curve to fit better the experimental data as the value of $1.4 \times 10^6\,\mathrm{m\,s^{-1}}$ is well-established and was obtained independently in specific experiments. Instead, we argue that our results offer a way to infer the value of $d_\perp$ from experiment, admittedly being a determination of the surface-response function for this particular wavelength only and for this particular interface between Au and $Al_2O_3$ [i.e., $d_\perp^{Al_2O_3-Au}(\omega_0)$]. In principle, this procedure amounts to a fitting-extraction of the dimensionless quantity $d_\perp/t_\mathrm{d}$, so that any inevitable uncertainties in $t_d$ (fabrication-induced deviations from nominal values) would in practice limit the accuracy by which we could in turn estimate $d_\perp$. Since theory accounts, including ab initio calculations, suggest that $d_\perp$ is to be found in the ångström range[45,59], we would need to experimentally determine $t_\mathrm{d}$ with atomic-scale accuracy, which remains a significant challenge as mentioned above. The fitting of the data with Eq. (1a) gives a value of $\delta_\mathrm{nl} = -0.0822 + 0.0303$ $\mathrm{i} \pm (0.007 + 0.02\mathrm{i})$ for the smallest studied dielectric gap $(t_\mathrm{d} = 2$ nm), which indeed meets our initial expectation that $|\delta_\mathrm{nl}| \ll 1$. While here we suggest a nonlocal interpretation for the observed gap-dependent broadening, we acknowledge that in principle this could also be qualitatively explained within the LRA by invoking other more lossy material-response models (see Supplementary Fig. S7), which phenomenologically mimic additional effects of roughness, grain boundaries, etc. While our experiments cannot totally dismiss such alternative explanations, they appear less obvious for our samples fabricated using high-quality materials. Indeed, SNOM measurements on single Au flakes[33,34]—where the bare surface plasmons exhibit the local-response dynamics, since there are no small spatial sample features to promote significant effects from the large wavevector response of the plasmons—appear well-explained by the Olmon data for crystalline Au[66], while more recent experiments conclude even lower bulk damping[82,83] in better accordance with the McPeak data[22].

In summary, we have performed an experimental study of record-high GSP mode-index in MDM structures comprised of high-quality monocrystalline Au flakes and ultrathin ALD-deposited $Al_2O_3$ films. Analysis of s-SNOM measurements from samples with different dielectric gap thicknesses, reveals signatures of gap-dependent nonclassical broadening, which we found to be well-described by the generalized hydrodynamic model of plasmonics[41]. Such broadening becomes progressively more significant for smaller dielectric gap thicknesses that give rise to higher effective-mode indices. Our results suggest that quantum nonlocal corrections should be taken into account when treating extremely confined gap plasmon modes supported by MDM structures with sub-10-nanometer dielectric gaps.

## Methods

**Sample fabrication.** The fabrication recipe flowchart diagram can be found in Supplementary Fig. S1. Below we provide details about particular fabrication steps and used equipment.

**Synthesis of monocrystalline Au flakes.** Monocrystalline Au flake samples were prepared using a recipe adopted from reference[24]. In short, thin and flat Au crystals were synthesized on BK-7 glass substrates via endothermic reduction of chlorouauric acid ($HAuCl_4 \cdot 3H_2O$) in ethylene glycol ($C_2H_6O_2$). All reagents were purchased from Sigma-Aldrich. The substrates were put into a vial with the solution and kept in an oven at 90 °C for 24 h. Afterwards, samples were cleaned in acetone, isopropyl alcohol (IPA) and distilled water and dried with a nitrogen blow. Each substrate carried a large number of Au flakes with diverse sizes and shapes, and suitable samples were selected by visual inspection using optical microscopy.

**FIB milling of waveguide couplers.** Milling of tapered waveguide coupler structures in monocrystalline Au flakes was performed using a gallium (Ga) ion FIB instrument (Helios NanoLab G3 UC, FEI company), with beam oriented perpendicularly to the sample plane an accelerating voltage of 30 kV and an ion current of 7.7 pA. In a preparatory step, substrates carrying Au flake samples that were selected in the previous fabrication step were coated with a thin ($\approx$6 nm) conductive carbon layer with a Leica Sputter Coater LEICA EM ACE600 in order to discharge the sample directly on the glass substrate, subsequently removed by mild plasma etching during the next fabrication step.

**ALD of dielectric gaps.** After FIB patterning, the samples were coated with 2, 3, 5, 10, and 20 nm–thick layers of $Al_2O_3$ using Oxford Plasma Technology OPAL ALD system (Bristol, UK) equipped with an inductively coupled plasma source. To avoid degradations of the samples, the aluminum oxide layer was deposited at low substrate temperatures (30 °C). As precursors trimethylaluminum (TMA) and oxygen plasma were used, with dose times of 30 ms and 5 s, respectively. Between each dose step, the reaction chamber was purged with inert argon and nitrogen for 5 s. The ALD process is defined by two half-cycles of self-terminating single surface reactions of each precursor, resulting in highly homogeneous and conformal $Al_2O_3$ coatings with nearly perfect thickness control[84]. The growth per cycle was 2.2 Å[85]. Hence, the desired film thickness was adjusted by the number of applied ALD-cycles.

**Transfer and assembly of Au flakes.** Upper Au flakes were transferred from the substrates at which they have been synthesized using customized 2D material transfer system (HQ+ graphene). Polydimethylsiloxane (PDMS) stamps (WF X4 Gel-Film from Gel-Pak) were used as a carrier substrates. Transfer was performed at elevated temperature of 130 °C to promote adhesion of the Au flake to the target substrate.

**Numerical electrodynamics simulations.** Numerical simulations and optimization of the tapered coupler geometric parameters were carried out using a commercially available finite-element method (FEM) solver (COMSOL Multiphysics 5.4, Wave Optics module). Since a planar waveguide was considered, modeling domain was two-dimensional (i.e., no spatial variation along $y$-axis, see Supplementary Fig. S13). Experimentally measured values of material optical parameters at 1550 nm wavelength were used: $\varepsilon_\mathrm{m} = -106.62 + 6.1257\mathrm{i}$ for monocrystalline Au from Olmon et al.[66]; $\varepsilon_\mathrm{d} = 2.657$ for $Al_2O_3$ from Boidin et al.[67]; and $\varepsilon = 2.1$ for the BK-7 glass.

Simulations were performed in a wavelength domain at $\lambda_0 = 1550$ nm using a Gaussian beam source in a scattered field formulation. Value of 0.15 was used as a numerical aperture of the Gaussian beam to mimic the experimental excitation conditions in the s-SNOM setup. Geometrical parameters of the tapered waveguide coupler ($w_\mathrm{t}$, $w_\mathrm{a}$, $w_1$ and $w_2$, see Supplementary Section S9) were optimized using the Levenberg–Marquardt algorithm (available from the COMSOL Optimization module), with the coupling efficiency being the objective function.

**Near-field measurements.** NF measurements were performed with the aid of s-SNOM[61], using the transmission module of a customized commercially available setup (Neaspec). Pseudo-heterodyne demodulation allows to simultaneously obtain information about amplitude and phase of the NF signal. Figure 3a shows a schematic diagram of the setup: the laser beam from a CW telecom laser (with wavelength $\lambda_0 = 1550$ nm) is split into two interferometric arms using a beam splitter (BS). In the reference arm, the signal is modulated using an oscillating mirror (OM) driven at the frequency $f \approx 300$ Hz. In the other arm, the laser beam is focused onto the waveguide coupler of the sample using a parabolic mirror (PM), with the full-width-half-max (FWHM) of the spot size ~3 $\mu$m. An atomic-force

microscope (AFM) tip (Arrow NCPt from NanoWorld) raster-scans the surface of the sample (which simultaneously allows to obtain topography of the sample) in a tapping mode (at the frequency $f_d \approx 250$ kHz and amplitude ~50 nm) and scatters the optical NF that is collected by another PM. Finally, reference and measurement arms are combined with a BS to allow interferometric detection and subsequent pseudo-heterodyne demodulation. In order to suppress background (bulk scattering from the tip and sample), the signal is demodulated at high harmonics of tip's oscillation frequency, $mf_d$, with $m = 3$ for the presented results. The recorded data for the propagation constant is reproducible using different, but nominally identical tips, while inevitable tip-to-tip variations would manifest locally in the intensity of scattered light and in the resolution of spatial dynamics below the tip radius. Details on the subsequent analysis of the recorded near-field maps is provided in Supplementary Section S4.

## Data availability

The data that underlie the findings of this study are available from the corresponding authors upon reasonable request.

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

## Acknowledgements

We are grateful to S. Raza and T. Christensen for their early theory contributions[48] that motivated this experimental study, and we also acknowledge stimulating discussions with C. Tserkezis. C.W. acknowledges funding from a MULTIPLY fellowship under the Marie Skłodowska-Curie COFUND Action (grant agreement No. 713694). S.I.B. acknowledges the support from VILLUM FONDEN (Villum Kann Rasmussen Award in Technical and Natural Sciences 2019). J.-S.H. acknowledges the support from Leibniz-IPHT (2020 Innovation Project) and DFG (HU 2626/3-1). N.A.M. is a VILLUM Investigator supported by VILLUM FONDEN (Grant No. 16498).

## Author contributions

S.B., J.-S.H., and N.A.M. conceived and planned the research. S.B. designed the MDM coupling structure, drawing also on expertise in MDM waveguides from V.A.Z. and S.I.B. P.A.D.G. and S.B. derived the dispersion relation within surface-response formalism. Samples were fabricated by S.B. and Z.-H.L., while S.B. synthesized the Au flakes and M.Z. performed the ALD. The FIB milling and SEM characterization was performed by S.B., Z.-H.L., and A.D. SNOM measurements were performed by V.A.Z. and S.B., while S.B., V.A.Z., and C.W. performed the data analysis. All authors contributed to the discussion and interpretation of the results, and the writing of the manuscript was done in a joint effort.

## Competing interests

The authors declare no competing interests.
