## [Peer review file · Nature Communications]

REVIEWER COMMENTS

Reviewer #1 (Remarks to the Author):

In the present work Boroviks et. al. report on the non-locality effect on gap plasmon modes. They show a series of optical experiments using s-NSOM in samples consisting of monocrystalline Au flakes forming a gap which is controlled with nanometric precision by means of the growth of a dielectric layer by ALD. Finally, the light coupler is formed using lithography.

Although this work is part of a series of works that this research group has been carrying out in recent years (as seen in the references), it contains interesting elements such as the combination of colloiddally grown flakes obtaining extensive regions of Au with low roughness (eventually atomic) and monocrystalline structure, allowing to dispel the presumption of artefacts and simplify the subsequent analysis of the NSOM images.

The main conclusion obtained: the retrieving of the nonlocality effects at different gap distances, is strongly supported by theoretical developments and experimental data treatment, it is found that the influence of non-local effects has to be considered when the gap is below 10nm. The greatest controversy could be considered the assumption that the deviations from the predicted behavior with respect to the obtained one, are attributed to the presence of "air voids", however, nothing suggests that the authors are not correct, and that leaves room for further improvements in the characterization of these type of structures.

I have a few comments and suggestions to be addressed:

- 1) How reproducible are the GSP parameters obtained when measuring the same System with different s-NSOM tips?
- 2) Is there some effect on the response regarding the thickness of the flakes used? e. i. a change of the effective dielectric constant of Au due to an extra electronic confinement.
- 3) What is the expected tolerance in the spatial homogeneity of the gap thickness? Could the authors add some additional calculations in that regard?
- 4) In figure 4b, the comparison between LRA and GNOR is made. Can the authors describe in a more extended way how the error bars reported for the experimental points are obtained?

Reviewer #2 (Remarks to the Author):

The work submitted by the authors reports on the experimental investigation of MIM waveguides with extremely confined gaps, with the goal of inferring the occurrence of nonlocal effects by the SNOM analysis of the complex propagation constant. This method is indeed very promising and has the perspective to provide a quantification of nonlocal effects without relying on any sub-nm-accurate morphological characterizations that are required when spectral shifts are instead analyzed. The paper is well written, sound, and the topic of investigation is timely. For all these reasons, I believe that the paper has the potential to be published on Nature Communications. However, as the authors also recognize, nonlocal effects are very subtle to demonstrate, even with the presented experimental approach, and one has to make any possible efforts to rule out possible competing explanations. For this reason, I believe that to provide a stronger evidence that their data point towards nonlocal corrections, and thus publish this message, the author should:

- Reinforce the statement at the end of the 'Discussion' session that 'more lossy material-response models' could also account for the observations but 'appear less obvious'. One possible way to do that would be e.g. to show in the supplementary information the experimental analysis of a system fabricated out of similar flakes that is not expected to display any nonlocal effect (e.g. a standard two-wire waveguide, a single rod, a gap antenna with large gap) and show that indeed its loss-related response (e.g. the Q factor of a resonance) is well reproduced by the same local dielectric function employed in the submitted paper.

- Further discuss the fact that they consistently get smaller real parts of the effective index compared to any model. The authors argue that this might be due to an underestimation of the thickness. However, in order to make the statement stronger, I would recommend to show that, when adjusting the thickness to better fit the real part of the index, the fitting of the imaginary part still remains accurate.

- Further discuss the parameters in the nonlocal correction, in particular if possible give any evidence from the literature that the extracted numbers for the diffusion constant and the carrier-scattering length have the correct order of magnitude.

Once these points have been addressed, I believe the paper can be considered for final publication. I will be happy to read again the manuscript if needed.

Response to the reviewer's comments

In the following, blue-colored font is used to designate quotations from the Reviewers' reports; black font is used for our responses; changes and additions to the manuscript are highlighted with a red color.

Reviewer #1:

In the present work Boroviks et. al. report on the non-locality effect on gap plasmon modes. They show a series of optical experiments using s-NSOM in samples consisting of monocrystalline Au flakes forming a gap which is controlled with nanometric precision by means of the growth of a dielectric layer by ALD. Finally, the light coupler is formed using lithography. Although this work is part of a series of works that this research group has been carrying out in recent years (as seen in the references), it contains interesting elements such as the combination of colloiddally grown flakes obtaining extensive regions of Au with low roughness (eventually atomic) and monocrystalline structure, allowing to dispel the presumption of artefacts and simplify the subsequent analysis of the NSOM images.

Response: We thank the Reviewer for consideration and positive assessment of our manuscript.

The main conclusion obtained: the retrieving of the nonlocality effects at different gap distances, is strongly supported by theoretical developments and experimental data treatment, it is found that the influence of non-local effects has to be considered when the gap is below 10nm. The greatest controversy could be considered the assumption that the deviations from the predicted behavior with respect to the obtained one, are attributed to the presence of "air voids", however, nothing suggests that the authors are not correct, and that leaves room for further improvements in the characterization of these type of structures.

Response: Indeed, we attribute the deviations in the experimentally measured effective mode index from the predictions of the GNOR model to the to presence of air voids and contamination of the samples. However, this does not affect our conclusion, but rather stands out the value of our method. This assumption is consistent with the observed trend: for a given nominal gap thickness t_d deviation is random, but experimentally obtained $\text{Re}\{n_{\text{GSP}}\}$ is always smaller than that predicted by theory. Nevertheless, our analysis shows that the need for the precise characterization of the gap thickness is to some degree relaxed when the results are considered in a parametric fashion. In order to elucidate this point and robustness of our method against deviations from the nominal gap thickness, we have added additional discussion and calculations in **Supplementary Section S8** and a remark on **p4**:

As such, the dashed curve in Fig. 4b depends only on ϵ_m and ϵ_d , while the solid curve has an additional dependence on δ_{nl} . The solid curve is swept away from the dashed counterpart when one increases δ_{nl} (effectively by varying complex-valued ξ , see inset in Fig. 1). Furthermore, as shown in Fig. S8, the slopes of both, LRA and GNOR dispersion curves in this parametric space are nearly independent of ϵ_d , as well as they are insensitive to addition of a thin air void in the gap (see Fig. S11). This relaxes the need for extremely accurate quantification of the possible deviations of ϵ_d and t_d (as fabricated) from its nominal values (as intended in the initial design).

We further discuss related aspects in our replies to question 3 and question 2 of the Reviewer #2.

1) How reproducible are the GSP parameters obtained when measuring the same System with different s-NSOM tips?

Response: We appreciate the reviewer's concern and reassure that we have verified the reproducibility of our measurements using using different (though nominally identical) commercially available tips. While the sharpness of the tip is undoubtedly relevant to the ultimate spatial resolution that one could achieve, we emphasize that this aspect does not affect our subsequent analysis. The explanation for this is, that the actual difference between the tips manifests in *a) intensity of scattered light* and *b) map resolution* (i.e., blurring of features that are smaller than the tip radius). These effects do not influence extraction of the GSP propagation constant: it is estimated from the normalized near-field amplitude and relative phase from different points of the same map, that is unaffected by the absolute scattering amplitude and subwavelength-scale blurring. As long as the oscillations at the frequency of interest can be spatially resolved, our Fourier analysis remains reliable even with less sharp tips. We now mention this point briefly in the **Methods Section**:

The recorded data for the propagation constant is reproducible using different, but nominally identical tips, while inevitable tip-to-tip variations would manifest locally in the intensity of scattered light and in the resolution of spatial dynamics below the tip radius.

2) Is there some effect on the response regarding the thickness of the flakes used? e. i. a change of the effective dielectric constant of Au due to an extra electronic confinement.

Response: We thank the Reviewer for bringing this aspect to discussion. In designing the experiment, our main emphasis has been on flake thicknesses that are sufficiently large that the realized structure mimics a metal–dielectric–metal (MDM) structure where the dielectric layer of finite thickness is effectively surrounded by semi-infinite metal regions. In linear regime, for extremely thin flakes, confinement effects could play a role when the thickness approaches a few Fermi wavelengths, as e.g. considered by Echarri *et al.*, *Optica* **6**, 630 (2019) and *Optica* **8**, 710 (2021). In our samples, the thickness of the gold flakes is safely beyond this regime, without any significant thickness dependence caused by electronic confinement. On the other hand, the electronic confinement associated with surface termination is naturally present for any flake thickness — this is exactly what is being accounted for through the surface-response formalism. In the revised manuscript we now briefly comment on this on **p3**:

... while the assignment of a local thickness-independent ε_m is also justified [57,62].

Another effect that is dependent on the thickness of the gold flakes is leaking of the gap mode, or in other words, hybridization of the MDM mode with DMD modes at neighboring interfaces due to the thinness of the metal layer. As mentioned on **p3** of the revised manuscript, characteristic length scale for this effect is skin depth (~ 30 nm), as we additionally confirm by numerical simulations, which are shown on the figure below. Here, effective mode index n_{GSP} is plotted as a function of thickness of the upper gold flake t_u for the indicated dielectric gap thicknesses and fixed thickness of the bottom gold flake t_b :

It can be clearly seen that for thicknesses $t_u \gtrsim 30$ nm, for all considered gap thicknesses t_d , effective mode index approaches limiting value. These results confirm that mode hybridization is not relevant for our samples, which have thickness of the upper flake ~ 50 nm and bottom flakes ~ 100 nm.

3) What is the expected tolerance in the spatial homogeneity of the gap thickness? Could the authors add some additional calculations in that regard?

Response: We thank the Reviewer for raising this important question. From our s-SNOM measurements we are aware of sample-to-sample variations and that, inevitably, the gap thickness varies slightly throughout samples. Apparently, it depends on various defects: air voids between the dielectric and transferred gold flake; surface imperfections and contamination of the bottom flake; contamination on dielectric layer, and of the eventually added top flake. All these defects increase the gap thickness and change the effective dielectric constant $\epsilon_{d,\text{eff}}$. In the revised manuscript, we emphasize that the increase in the gap thickness does not invalidate our subsequent analysis. We have added discussion and new calculations in Supplementary Section S8, confirming that while addition of an air layer in the gap decreases the real part of n_{GSP} , it does not significantly change the slope of the dispersion curve in parametric representation. Similar conclusion can be drawn from supplementary Fig. S8: even significant change of the dielectric constant ϵ_d changes the slope of the parametric curve insignificantly, but rather almost proportionally scales both real and imaginary parts of n_{GSP} . Besides, based on our assumption of an additional air void, we estimate its thickness t_{air} as presented in Supplementary Fig. S12. Our calculations show that t_{air} varies from sample to sample (~ 0.3 to ~ 4 nm). Thus, the tolerances in the spatial homogeneity of aluminum oxide layer appear negligible compared to the above issues: ALD offers controlled growth of homogeneous dielectric layers with approximately ~ 0.2 nm precision [82].

Besides, plausible inhomogeneities of both t_d and $\epsilon_{d,\text{eff}}$ within a given sample, distort the oscillatory pattern of the propagating GSP modes and consequently contribute to broadening of the peaks in the averaged k_x spectrum. In turn, this results in the increased widths of the horizontal errorbars in Fig. 4b. Fig. S3d also shows that widths of the peaks corresponding to n_{GSP} indeed vary from sample to sample, pointing to presence of the gap inhomogeneities. As we have already mentioned above and as was addressed in the original manuscript, our experimental procedure allows us to extract the complex-valued δ_{nl} with a relative accuracy on the order of

10 percent for the real part [$\delta_{nl} = -0.0822 + 0.0303i \pm (0.007 + 0.02i)$] irrespective of the detailed uncertainties in the gap, while any subsequent estimation of the associated surface-response function d_{\perp} would indeed require stricter tolerances on our knowledge of the actual gap thickness. A further related discussion can be found in our reply to the question 3 of the Reviewer #2.

4) In figure 4b, the comparison between LRA and GNOR is made. Can the authors describe in a more extended way how the error bars reported for the experimental points are obtained?

Response: We thank the Reviewer for bringing the procedure for the error bars to discussion. Indeed, a section Supplementary Information (now S4 in the revised manuscript) offers exactly these details. In the revised main text we now explicitly refer to the SI: **Further details about NF map processing can be found in Supplementary Section S4.**

Reviewer #2:

The work submitted by the authors reports on the experimental investigation of MIM waveguides with extremely confined gaps, with the goal of inferring the occurrence of nonlocal effects by the SNOM analysis of the complex propagation constant. This method is indeed very promising and has the perspective to provide a quantification of nonlocal effects without relying on any sub-nm-accurate morphological characterizations that are required when spectral shifts are instead analyzed. The paper is well written, sound, and the topic of investigation is timely. For all these reasons, I believe that the paper has the potential to be published on Nature Communications. However, as the authors also recognize, nonlocal effects are very subtle to demonstrate, even with the presented experimental approach, and one has to make any possible efforts to rule out possible competing explanations. For this reason, I believe that to provide a stronger evidence that their data point towards nonlocal corrections, and thus publish this message, the author should:

Response: We appreciate the Reviewer's high opinion on our work and thank them for recommending publication in Nature Communications and suggesting improvements.

1) Reinforce the statement at the end of the 'Discussion' session that 'more lossy material-response models' could also account for the observations but 'appear less obvious'. One possible way to do that would be e.g. to show in the supplementary information the experimental analysis of a system fabricated out of similar flakes that is not expected to display any nonlocal effect (e.g. a standard two-wire waveguide, a single rod, a gap antenna with large gap) and show that indeed its loss-related response (e.g. the Q factor of a resonance) is well reproduced by the same local dielectric function employed in the submitted paper.

Response: We thank the Reviewer for encouraging such an important discussion.

First, we note that largest of the considered gap thicknesses ($t_d = 10, 20$ nm) are already too large to promote any significant nonlocal effects: in this regime nonlocal dispersion asymptotically approaches LRA dispersion. This is indeed what we

observe for the experimental data points in the lower left-hand corner of Fig. 4b. In this regime, measured n_{GSP} are well explained by the local description using a local dielectric function for the crystalline gold and the dielectric layer.

Further along the lines encouraged by the Reviewer, in our recent work, Lebsir *et al.* arXiv:2203.00754, we report s-SNOM measurements on single flakes where the surface plasmons exhibit the local-response dynamics, since there are no small spatial sample features to promote effects from the large wavevector response of the plasmons. We now reference this work in the revised manuscript, along with similar studies by Kaltenecker *et al.* that were also referenced already in the original manuscript,] as well as recently appeared pre-print by Casses *et al.* In fact, all of the mentioned works conclude that the bulk material losses in monocrystalline gold flakes are even lower than expected from Olmon's reported permittivity. We have added a comment on p5: Indeed, SNOM measurements on single Au flakes [33,34]—where the bare surface plasmons exhibit the local-response dynamics, since there are no small spatial sample features to promote significant effects from the large wavevector response of the plasmons—appear well-explained by the Olmon data for crystalline Au [64], while more recent experiments conclude even lower bulk damping [80,81] in even better accordance with the McPeak data [22].

2) Further discuss the fact that they consistently get smaller real parts of the effective index compared to any model. The authors argue that this might be due to an underestimation of the thickness. However, in order to make the statement stronger, I would recommend to show that, when adjusting the thickness to better fit the real part of the index, the fitting of the imaginary part still remains accurate.

Response: We recognize the Reviewer's concern and thank him/her for suggesting an argumentation to elucidate our statement.

Indeed, as was initially spelled in the original manuscript, we do find consistently smaller real parts of the effective index compared to expectations based on the nominal gap sizes. One likely explanation is that the actual gap sizes exceed the corresponding nominal values, e.g., associated with unintended air voids/contamination layers. Alternatively, there could be a modification of dielectric function of the aluminum oxide layer, which is though quite unlikely, given that ALD is well-established and reproducible method. We have already shown in the supplementary information of the original manuscript (now Fig. S8) that variation of the effective dielectric function has only modest effect on the parametric dispersion curve, while points on this parametric curve move along the curve as one parametrically varies the thickness (i.e. the slope of the parametric curves is nearly independent of the effective dielectric function). This same behavior is observed for corresponding nonlocal parametric curve.

In the revised manuscript, we have added **Supplementary Section S8** that presents discussion and additional calculations concerning the influence of potentially present air void on the dispersion relation. As shown in Fig. S12, we find that increasing the gap thickness t_d and accordingly adjusting $\epsilon_{d,\text{eff}}$ allows to fit both, real and imaginary parts of the effective mode index. A related discussion can be also found above, in our reply to question 3 of the Reviewer #1.

3) Further discuss the parameters in the nonlocal correction, in particular if possible give any evidence from the literature that the extracted numbers for the diffusion

constant and the carrier-scattering length have the correct order of magnitude.

Response: We thank the Reviewer for bringing this important point to the discussion. Perhaps challenging common expectations, we emphasize that literature with accurate tabulation of experimental nonlocal corrections remains missing, which was the main motivation underlying the current study. Nevertheless, we have added an overview of the previously reported values (both, experimental and theoretical) of the diffusion constant \mathcal{D} , Table S1 in Supplementary Section S5 and a paragraph on p4 of the main text:

The diffusion constant \mathcal{D} is used as the GNOR model's fitting parameter (see details in Supplementary Section S5). By fitting the dispersion relation [Eq. (1a)] to the imaginary part of the experimental data points (at the corresponding values of $\text{Re}(n_{\text{GSP}})$), estimated $\mathcal{D} \simeq 8 \times 10^{-4} \text{ m}^2 \text{ s}^{-1}$. We emphasize that this procedure does not strictly rely on particular assumptions about the parameters of the dielectric gap, thus allowing a robust extraction of \mathcal{D} even with significant uncertainties in the thickness t_d and effective permittivity ϵ_d . Further justification of this point is discussed in the following section.

In the initial GNOR theory, naturally, there were some estimates based on the bulk scattering time [*Nat. Commun.* **5**, 3809 (2014)], while later it was consolidated that the diffusion is effectively linked to the surface-scattering properties [*J. Phys. Cond. Matter* **32**, 395702 (2020) & *Nanophotonics* **10**, 2563 (2021)]. That being said, based on Kreibig broadening, the diffusion constant that mimics Landau damping (at the surface between gold and air) is anticipated to be of the order $\mathcal{D} \sim 3 \times 10^{-4} \text{ m}^2 \text{ s}^{-1}$ [*J. Phys. Cond. Matter* **32**, 395702 (2020)], while potentially increasing in any additional presence of also surface-roughness induced scattering. The magnitude of our experimentally extracted diffusion constant — $\mathcal{D} \sim 8 \times 10^{-4} \text{ m}^2 \text{ s}^{-1}$ — potentially indicates some additional surface scattering in our samples, despite our dedicated efforts to eliminate this through our exploitation of crystalline materials and ALD layers. Alternatively, the dielectric layer (rather than air) could also promote slightly more Landau damping [*Phys. Rev. Lett.* **115**, 193901 (2015)], which could again manifest in a slightly increased surface-scattering diffusion constant. In the revised manuscript, we now mention these possibilities on p4:

The diffusion constant \mathcal{D} , which represents the carrier scattering, was estimated from a fit to the experimental data (see Section S5 of the Supplementary Information). Obtained value $\mathcal{D} \simeq 8 \times 10^{-4} \text{ m}^2 \text{ s}^{-1}$ is slightly smaller than previously reported experimental value $\mathcal{D} \simeq 8.8 \times 10^{-4} \text{ m}^2 \text{ s}^{-1}$ [73]. In the initial GNOR theory [41], it was consolidated that the diffusion is effectively linked to the surface-scattering properties [76] and the diffusion constant that mimics Landau damping (at the surface between gold and air) is anticipated to be even smaller, on the order of $\mathcal{D} \sim 3 \times 10^{-4} \text{ m}^2 \text{ s}^{-1}$ [77], while raised dielectric surroundings ($\epsilon_d > 1$) could also promote further Landau damping [78,79]. Eventually, the higher diffusion constant revealed by our experiments (as well as Ref. 73) could also indicate some additional surface scattering in our samples, despite our dedicated efforts to eliminate roughness through our exploitation of crystalline materials and ALD layers. Overview and comparison of the previously reported values of \mathcal{D} is provided in Section S5 of the Supplementary Information.

REVIEWERS' COMMENTS

Reviewer #1 (Remarks to the Author):

I thank the authors for considering the suggestions made, and for making changes to the article text and the SI accordingly. I recommend its publication in Nature Communication.

Reviewer #2 (Remarks to the Author):

I stand with my previous report in considering the results interesting, timely and deserving publication. In this rebuttal, I find that the authors have thoroughly and successfully addressed all the open issues raised both by the other reviewer and by myself. In particular, I believe that they properly consolidated the conclusion that their results strongly point towards nonlocality and that their analysis to exclude other possible origins for the observed behavior is solid. In the present form, I am in favour of publication in Nature Comm without further changes.